# RNA-Based Liquid Biopsy in Head and Neck Cancer

**DOI:** 10.3390/cells12141916

**Published:** 2023-07-23

**Authors:** Jacek Kabzinski, Aleksandra Kucharska-Lusina, Ireneusz Majsterek

**Affiliations:** Department of Clinical Chemistry and Biochemistry, Medical University of Lodz, MolecoLAB A6, Mazowiecka 5, 92-215 Lodz, Poland; jacek.kabzinski@umed.pl (J.K.); ola_kucharska@wp.pl (A.K.-L.)

**Keywords:** cancer, HNC, liquid biopsy, mRNA, miRNA, circRNA, lncRNA, piRNA, snRNA, snoRNA

## Abstract

Head and neck cancer (HNC) is a prevalent and diverse group of malignancies with substantial morbidity and mortality rates. Early detection and monitoring of HNC are crucial for improving patient outcomes. Liquid biopsy, a non-invasive diagnostic approach, has emerged as a promising tool for cancer detection and monitoring. In this article, we review the application of RNA-based liquid biopsy in HNC. Various types of RNA, including messenger RNA (mRNA), microRNA (miRNA), long non-coding RNA (lncRNA), small nuclear RNA (snRNA), small nucleolar RNA (snoRNA), circular RNA (circRNA) and PIWI-interacting RNA (piRNA), are explored as potential biomarkers in HNC liquid-based diagnostics. The roles of RNAs in HNC diagnosis, metastasis, tumor resistance to radio and chemotherapy, and overall prognosis are discussed. RNA-based liquid biopsy holds great promise for the early detection, prognosis, and personalized treatment of HNC. Further research and validation are necessary to translate these findings into clinical practice and improve patient outcomes.

## 1. Head and Neck Cancer

Cancer is one of the greatest challenges of modern medicine. Despite the huge funds allocated to research, prevention and treatment, both the number of cases and mortality increases every year [1]. Head and neck cancers (HNC) are part of this trend and rank seventh in the world in terms of new cases (660,000) and deaths (325,000) per year [2,3]. HNC is a wide group of diseases, and while there is a lack of consensus on the definitions of ‘head and neck cancer’ terminology, according to GLOBOCAN they include primarily: lip cancer, oral cavity cancer, larynx, nasopharynx, oropharynx and hypopharynx cancers [1]. A detailed breakdown of HNC by location is presented in Figure 1.

The increase in morbidity and mortality is driven mainly by oropharyngeal cancer, and the main causative factors are aging and growth of the population as well as changes in the prevalence and distribution of the main environmental risk factors such as smoking, alcohol consumption (and in particular both factors accounting for 72% of cases when used in combination [4]) and human papillomavirus (HPV) infection (especially HPV type 16, a major risk factor for oropharyngeal cancer [5]) [3,6]. Moreover, risk factors may interact with each other, leading to a cumulative increase in risk [7]. Other factors such as diet [8,9], genetics [10] and oral hygiene [11,12] are considered to have a minor impact on risk modulation.

It is estimated that the annual increase in the incidence of HNC may reach 30% per year until 2030, and the decreasing number of HPV-unrelated cases in Europe and the USA caused by the reduction in smoking and the decline in the consumption of strong alcohols will be compensated by the increase in HPV-related cases (mainly associated with oropharyngeal cancer). At the same time, HPV-unrelated cases are expected to rise in the Asian region [13]. This will probably mean a change in the number of individual HNC types, leading to a situation in which oropharyngeal cancer incidence will overtake cancer of the oral cavity [14].

In terms of survival, HNC patients generally have a poor prognosis. The five-year survival rate varies between 30 and 70%, depending on the stage and location of the tumor [15], and the average value for all variants is 50% [16]. These rates have remained at a similar level for decades, despite the development of treatment methods and the emphasis on early diagnosis [17]. Most HNC cases begin in the squamous cells giving rise to squamous cell carcinoma (HNSCC), which accounts for approximately 90% of all head and neck cancers [2,18].

All the above-mentioned, not very optimistic, statistics, as in the case of most other types of cancer, have created an urgent need to find novel diagnostic methods for HNC. The basic diagnostic path for head and neck cancers includes primarily imaging diagnostics (endoscopy, head MRI, CT of the sinuses and head, panoramic dental x-ray, dental cone beam CT, PET/CT) supported by histopathological tests [19]. At the same time, the pursuit of faster, more effective and less invasive diagnostic methods leads to the introduction of innovative tests such as immunochemistry [20,21], molecular analyses [22,23,24], ultrasound elastography [25,26] or Raman spectroscopy [27]. There are even reports of the use of Artificial Intelligence-based methods in head and neck cancer diagnosis [28]. All these methods, although very promising, still require evaluation of effectiveness and long-term analysis in terms of usefulness in diagnostics, especially in terms of accuracy, ease of use, cost and invasiveness [29]. One of the particularly promising techniques in cancer diagnosis seems to be liquid biopsy, which, if only its effectiveness and specificity are proven, satisfactorily meets the requirements for an ideal biomarker—it is fast, cost-effective and minimally invasive.

## 2. Liquid Biopsy

The concept of liquid biopsy is gaining more and more recognition in the field of cancer diagnosis. Since from the very beginning of the neoplastic transformation of the cell, it secretes specific biomarkers, as well as the body itself produces specific compounds in response to the presence of cancer cells, the possibility of identifying and detecting such molecules in a phase where any imaging diagnostics is impossible seems to be particularly tempting as an element of ultra-early detection of cancer [30,31,32]. However, at the very beginning of considerations about liquid biopsy, it is necessary to mention the limitations of this method, resulting primarily from the early stage of its implementation and resulting in the lack of unification and standardization of procedures [33]. Nevertheless, such an implementation path applies to all innovative methods in medicine, hence the need for further research in order to fully implement and recognize liquid biopsy. This method is based primarily on the analysis of circulating tumor cells (CTCs), circulating tumor DNA (ctDNA), circulating different types of RNAs and tumor-derived extracellular vesicles (TD-EVs).

The history of liquid biopsy, although the term did not exist at the time, began in the early 2000s when it was noted that the expression profile of microRNA (miRNA) in patients with chronic lymphocytic leukemia was different from that of healthy individuals [34]. Since then, the popularity of research on and attempts to implement liquid biopsy has skyrocketed, as researchers have recognized three basic and undeniable benefits of such a diagnostic method. First, the biomarker is synthesized quickly in response to a pathological event. Second, it is highly specific. And finally, third, it remains in the system for a long time and is easily detectable due to its presence in the plasma. It should also be mentioned that, despite the term “biopsy”, the use of this method is not limited to cancer. The use of primarily miRNA has also found its application in the diagnosis of cardiovascular [35,36,37] and infectious diseases [38,39]. Nevertheless, the greatest amount of research using the widest spectrum of biomarkers is conducted in the field of using this method in the diagnosis of cancer [40,41,42], including HNCs [43,44,45], and the greatest hopes are placed in this project.

### 2.1. Circulating Tumor Cells

Circulating Tumor Cells (CTCs) are epithelial cells circulating in the bloodstream and their origin is a primary tumor. CTCs obtain genetic heterogeneity and undergo epithelial-mesenchymal transition (EMT) enabling migration and invasive potential [46]. Detection of CTCs in the bloodstream is the basis for prognosis of the severity of the disease, the metastatic potential of cancer and an indicator of the effectiveness of therapy [47]. This method, however, requires special techniques (concentration and enrichment), mainly due to the extreme rarity of CTCs, which usually range in number from about one in a billion blood cells [48]. So far, the analysis of circulating tumor cells has shown limited use in HNC diagnostics, and the detection of CTCs has been associated primarily with an advanced tumor stage [49] and disease progression rate and poor prognosis [50]. This method, while undeniably effective, is applicable only to more advanced stages of cancer, and its future use is likely to focus on metastatic potential [51].

### 2.2. Circulating Tumor DNA

Circulating tumor DNA (ctDNA) is a small fragment of tumor-derived DNA, usually about 150 to 200 base pairs in length, present in the blood of cancer patients. Its origin is apoptotic and necrotic tumor cells, and it carries somatic mutations of primary and secondary tumors. The detection of gene mutations in ctDNA, as well as epigenetic changes such as methylation pattern, allows for the usage of this biomarker as a convenient factor for diagnosis, prognostication and monitoring the response to therapy. This method is particularly useful in the case of large solid tumors and in the case of metastasis, where the concentration of ctDNA in the bloodstream increases significantly, facilitating the diagnostic process. ctDNA analysis has proven its usefulness by being approved by the FDA for monitoring non-small cell lung cancer [52] and as a screening marker for colorectal cancer [53]. In HNC the most frequently examined gene mutations are TP53, PIK3CA, CDKN2A, CASP8 and NOTCH1, and the incidence of ctDNA mutations with at least one gene mutated ranges from 42% to 88% [54]. All available data suggest that individualized ctDNA monitoring might be a promising biomarker for HNC, although further research is definitely needed [55,56,57].

### 2.3. Tumor Derived Extracellular Vesicles

Tumor-derived extracellular vesicles (TD-EVs) are heterogeneous small membranous vesicles spread from the surface of cells. They are used for intercellular communication acting as mediators [58] and are suspected of allowing cells to exchange proteins, lipids and genetic material [59]. Currently identified as TD-EVs are exosomes, nanovesicles, microvesicles, exomeres, apoptotic bodies and large oncosomes [60,61]. The main challenge in the analysis of TD-EVs is, as in the case of CTCs, the very low concentration in the bloodstream and the resulting need for sample enrichment [62], as well as technical problems during isolation due to the high risk of lipoprotein contamination [63]. In addition, the aforementioned composition of TD-EVs, which may consist of membrane-associated proteins and lipids, nucleic acids (especially RNAs), non-membrane lipids and proteins present in the cytosol or released by intracellular organelles [64], on the one hand, allows for the prospects of a broad multifactorial analysis but on the other hand requires particular caution and an unquestionable link between the examined factors and a specific pathological condition. However, if these challenges are overcome, a very valuable biomarker is obtained as a result, useful primarily in assessing the metastatic potential and stage of cancer, including head and neck cancers [63,65,66,67].

### 2.4. Metabolomic Markers

Metabolism-related biomarkers have received a great deal of attention in recent years. Metabolism affects various biological processes, including cell growth, proliferation, and energy production, and plays a critical role in cancer development and progression. Alterations in metabolic pathways can lead to the accumulation of specific metabolites or alter the activity of metabolic enzymes, which may serve as potential biomarkers for several types of cancer, including HNC. At the same time, the rapid development of analytical techniques used in metabolite analysis, mainly Gas Chromatography-Mass Spectrometry (GC-MS) used to detect compounds in biological material such as blood, plasma, saliva, urine or tissues, and Positron Emission Tomography (PET) for real-time visualization of, for example, glucose metabolism in tumors, are of great benefit and help in diagnosis, staging and treatment response assessment, while at the same time being part of the trend towards liquid biopsy, where speed, simplicity and non-invasiveness are as important as the diagnosis itself.

The first class of metabolism-associated biomarkers are metabolites. Metabolomic profiling studies have revealed distinct metabolic alterations in HNC patients compared to healthy individuals, providing valuable insights into the disease biology and potential diagnostic and prognostic markers. The most common metabolic alteration observed in HNC is the Warburg effect, characterized by increased glycolysis and lactate production even in the presence of oxygen. This metabolic reprogramming allows cancer cells to meet their energy demands and support rapid proliferation [68,69,70]. Lactate also accumulates in the tumor microenvironment due to enhanced glycolysis. Increased lactate levels in HNC tissues and body fluids, such as blood or saliva, have been observed and are associated with tumor aggressiveness, poor prognosis and treatment resistance [71,72]. Besides metabolites associated with glucose metabolism, another group of compounds linked to HNC is amino acids. Alterations in amino acid metabolism are observed in cancer cells to support their increased protein synthesis and provide metabolic intermediates for various biosynthetic pathways. Increased levels of glutamate, glutamine, methionine and serine have been reported in HNC, as well as changes in the ratios of specific amino acids, such as an increased glutamine/glutamate ratio, have been associated with tumor aggressiveness [73,74,75]. Lipid metabolism is also dysregulated in HNC, with changes in lipid composition and metabolism influencing cancer cell growth and signaling pathways. Alterations in fatty acid metabolism and lipid profiles have been reported in HNC patients, for example, phosphatidylcholine [76] or phosphatidylethanolamine [77] levels. Finally, there are reports on nucleotide levels dysregulation in HNC, which can lead to imbalances in nucleotide pools, impacting cancer cell growth. Studies have identified changes in purine and pyrimidine metabolites in HNC, suggesting their potential as biomarkers for disease progression and treatment response [70,78].

A second class of intensively studied compounds as metabolic biomarkers are metabolic enzymes and transporters. Dysregulation of key enzymes involved in metabolism, mainly related to the compounds mentioned above, has been observed in HNC. Hexokinase, pyruvate kinase and lactate dehydrogenase expression levels or activity can be evaluated as potential biomarkers to assess metabolic alterations in tumor cells [79,80,81]. Furthermore, altered expression of metabolic transporters, involved in the delivery or removal of the analyzed metabolites from the cell, such as glucose transporters (GLUTs) and monocarboxylate transporters (MCTs), has been linked to HNC development and progression, providing additional targets for biomarker investigation [70,82,83].

Metabolism-associated biomarkers in HNC, including metabolites, metabolic enzymes, transporters and imaging techniques, hold promise for improving diagnosis, prognosis and treatment monitoring. Their potential to reflect the metabolic alterations associated with HNC provides opportunities for personalized approaches and the development of targeted therapies. Further research is needed to validate these biomarkers, optimize their detection methods and explore their clinical utility in guiding HNC management, especially given the incredible complexity of metabolic pathways in the cell, which means that metabolic biomarker candidates are being analyzed by the hundreds [84,85], while thousands more are waiting to be studied.

## 3. RNAs in Blood-Based Diagnostics

Of all the previously mentioned cells and molecules used in liquid biopsy, RNA-based ones are a unique subgroup. Despite the fact that they were not the first to be discovered, they quickly gained the status of the most promising, and the majority of research has focused on their analysis. The increase in popularity of RNA-based biomarkers can be seen in factors that make it much more convenient to use than cancer DNA or cancer cells. Firstly, RNAs have much greater stability in the bloodstream compared to DNA or cells, which allows them to be detected in a much wider time window [86,87]. The negligible degradation over time makes RNAs not only non-tissue-specific markers but also non-time-specific ones, which is very desirable in the diagnostic process [88,89]. Secondly, the procedures for isolating and obtaining RNA molecules for analysis from the bloodstream are much faster and much simpler than in the case of DNA or whole cells, which often require complicated purification, enrichment and concentration procedures [48,87,90]. Thirdly, cancer cells release RNAs throughout their life cycle and not only in the case of apoptosis or metastasis, as is the case with the markers described earlier, which makes them useful in early diagnosis, and not only in the assessment of metastatic potential [91,92,93,94]. All these advantages, combined with the successes already achieved in the field of using RNAs as a biomarker, make it the most promising element in the field of diagnostics based on liquid biopsy.

### Types of RNA in Blood

All cells, including cancer cells, continuously release a range of different types of RNA, both intentionally (e.g., secreted for cell signaling) and unintentionally (e.g., as a result of apoptotic cell breakdown) [95,96,97]. Subtypes of RNA circulating in the blood that have so far been used in diagnostics, or are considered potentially useful in this area, include mRNA, microRNA (miRNA), long non-coding RNA (lncRNA), small nuclear RNA (snRNA), small nucleolar RNA (snoRNA), circular RNA (circRNA) and piwi-interacting RNA (piRNA). Their summary is presented in Table 1.

The continuous release of different types of RNA by cells at each stage of their life cycle means that, unlike CTCs and ctDNA, RNA can not only be a marker of apoptotic/necrotic events and metastasis but also, and perhaps above all, provide information about active cancer cells, including those at a very early stage of their development [100,104]. RNA can be released both in its entirety as a functional molecule and in the form of fragments after prior degradation. There is no universal agreement in the literature on the meaning of the commonly used term cf-RNA—depending on the source, it refers to either cell-free-RNA or circulating-free-RNA. Martinez-Dominguez et al. [105] postulates that the term cell-free-RNA should be used to classify the RNA released by the cell into the bloodstream and circulating-free-RNA to characterize the degraded fragments that have been revealed, e.g., during cell disintegration. However, despite the ambiguity in the nomenclature, both types of cf-RNA are used in diagnostics. In addition, classification based on size is often used, where the boundary between short RNA and long RNA is taken as 200 nucleotides. Obtaining RNA for analysis is relatively non-invasive for the patient and most often comes down to blood collection and the use of plasma or serum, although saliva and urine are also used as a source [107,108,109,110]. RNA obtained from biological material, depending on the type, differs significantly in stability, which largely determines the possibility of its effective use. mRNA is the least stable, while at the other end of the spectrum are microRNA and long non-coding RNA [86,111]. A summary of the features of RNAs circulating in the blood is provided in Figure 2.

## 4. RNAs in HNC Liquid-Based Diagnostics

### 4.1. mRNA

Messenger RNA is the best known and tested RNA type, and its functions and properties compared to other kinds of RNA seem to be relatively simple and understandable. This, and the fact that it was discovered long before other groups of RNA, meant that together with the aforementioned CTCs and ctDNA it was at the center of interest in the initial stages of research that we today call a liquid biopsy. The journey of mRNA-based molecular diagnostics of cancer began in 1999 along with the publishing by Golub et al. on classification and prediction methods by gene expression monitoring in myeloid leukemia and acute lymphoblastic leukemia [112]. However, with the increase in the awareness of the restrictions that these methods have in diagnostics, and after discovering new types of RNA, eyes and hopes turned to miRNA, lncRNA and other more promising molecules. This does not mean, however, that mRNA has been abandoned and recognized as useless; by respecting the restrictions and knowing what aspects of mRNA can be useful, we receive a very powerful diagnostic tool.

mRNA is a link connecting the information contained in DNA and a protein built based on this information; its role and function is, therefore, simple and does not change significantly in cancer cells [113]. However, since messenger RNA is a reflection of the level of gene expression in the cell, and this level undergoes significant dysregulation in cancer cells [114,115], mRNA can be used as an intermediate marker of abnormal processes occurring in the cell. The biggest limitation of the use of mRNA in liquid biopsy is the fact that cancer cells, including HNC cells, do not release mRNA outside the cell by default. A number of studies indicate the potential use of mRNA as a biomarker in HNC in the carcinogenesis process [116,117], predictions of recurrence or metastasis [118,119,120,121], outcome prediction [122,123], molecular classification of subtypes [124,125] and response for therapy prognosis [126,127,128]. However, all these procedures, due to the negligible secretion of the mRNA by cancer tissue, require “standard” biopsies and obtaining a tumor bioptate, which is a much more invasive method than taking blood or other body fluid and does not fit into the idea of liquid biopsy. However, there is a very good reason not to exclude mRNA from the group of particles classified as potentially useful in liquid biopsy—saliva analysis. Collection of a saliva sample is even less invasive than blood drawing, and numerous literature reports in the field of mRNA analysis in such samples seem to be promising. Bu et al. proved that increased expression of tissue and salivary transgelin mRNA can be used to predict poor prognosis in Oral Squamous Cell Carcinoma (OSCC) [129]. The analysis was performed not only in saliva but also in the plasma of patients, which makes this study one of the very few examples of the use of mRNA in the liquid biopsy of the blood-based material. Chai et al. propose that the detection of HPV-16 mRNA in salivary oral rinse is indicative of HPV status in HNSCC patients and can potentially be used as a diagnostic tool [130]. Oh et al. studied 30 candidate genes with relevance to HNC in whole saliva and found that the specific monoamine oxidase B (MAOB) and NGFI-A binding protein 2 (NAB2) expression pattern is a potent biomarker for early OSCC diagnosis especially in the group under 60 years of age [131]. Li et al. [132] analyzed transcripts of IL8, IL1B, DUSP1, HA3, OAZ1, S100P and SAT and found that the combinations of these biomarkers yielded sensitivity (91%) and specificity (91%) in distinguishing OSCC from the controls, which was later confirmed by Elashoff et al. [133]. The enormity of the number of genes whose expression is diverged in HNC means that mRNA analysis in saliva samples seems to be very promising and future-oriented in diagnostics. Particular attention is devoted to cytokines, metalloproteinases and proteins of acute phase, and the growing number of reports in this area provides mRNA a spot in a group of recognized markers in liquid biopsy, despite significant restrictions on the study of blood-based material [134]. In addition, reports on mRNA and lncRNA co-profiling in HNSCC indicate that maybe messenger RNA will play an additional, currently unknown role in a liquid biopsy in the future [135].

### 4.2. miRNA

In recent years, microRNA has been at the forefront of the most promising disease markers. The unique combination of properties including very high specificity, presence in the bloodstream and long lifetime promises a near-perfect biomarker. Particularly high intensity of research in this area can be observed in the case of cardiovascular diseases and cancer. The former still do not have a gold standard in the field of diagnostics, and the markers currently used are characterized by either low specificity, a very narrow time window of availability or local occurrence within specific tissues [37,136,137]. In the case of cancer, research is driven by the pressing need to develop markers for early diagnosis, prediction of treatment outcome by subtype and overall prognosis. In the case of cancer, most research is directed towards developing markers that allow early diagnosis as well as prediction of treatment outcomes and overall prognosis [138,139,140]. The pursuit of such markers for both cancer and CVD is undoubtedly due to the fact that these are currently the two most deadly diseases affecting humanity [141]. However, despite the intensity of research and the enormity of resources and funds allocated for this purpose, we still have not reached the point where miRNA could be considered a fully reliable biomarker ready for large-scale clinical use. Meanwhile, research is ongoing, and the results allow us to believe that we are close to the goal. At the beginning of 2023, 52 clinical trials using miRNA are registered in the European Union, of which 28 are still in progress [142]. At the same time, the US National Institute of Health reports on the ClinicalTrials.gov website about 1215 similar studies, of which 4 are devoted to head and neck cancers [143]. Detailed information is presented in Table 2.

The abundance of miRNA types and their high specificity in response to almost every event occurring in cells allows an attempt to identify biomarkers in a much wider range than in the case of CTCs, ctDNA or mRNA. The conducted studies for HNC distinguish miRNAs specific to the process of diagnosis, metastasis, tumor resistance to radio and chemotherapy treatment and general prognosis of the disease outcome.

#### 4.2.1. miRNA in HNC Diagnosis

The need for accurate tumor type determination is particularly pronounced in the case of HNC due to the extremely wide spectrum of heterogeneity in this group. The variety of subtypes entails a number of different therapeutic strategies, and the selection of the appropriate one will determine the success or failure of the treatment process. Malignant transformation causes changes in microRNA expression that can be tracked and used in head and neck cancer diagnosis [148,149]. Changes in expression may include downregulation, in which case miRNAs could function as tumor suppressors by negatively regulating oncogenes [150], or overexpression, where miRNAs could function as oncogenes (called oncomiRs) by inhibiting tumor suppressor genes [151]. The best described and most thoroughly studied oncomiR in HNC diagnostics is miR-21, in the case of which expression disorders are associated with dysfunction of gene expression, transport, signal transduction and degradation mechanisms within the cell [152]. Target genes for miR-21 are PTEN and PDCD4 [153]. Other oncomiRs with diagnostic potential include miR-134 (target gene PDCD7) [154], miR-4295 (NPTX1) [155], miR-501-5p (CLCA4) [156], miR-125a (TP53 and ERBB3) [157], mir762 (FOXO4) [158] and miR-196b (PCDH-17) [159]. The tumor suppressor miR group includes let-7 family (Ras) [160], miR-625 (SOX4) [161], miR-200 family (QKI) [162], miR-125b (PRXL2A) [162], miR-124 (STAT3) [163] and miR-140-5p (PAK4) [164]. All these miRNAs take part in the tumorigenesis process in HNC, and despite their high specificity, it has still not been possible to select one that would be universal enough to be implemented in clinical practice. A potential solution to this problem seems to be the creation of multi-miRNA panels, where a set of selected microRNAs allows for an unambiguous diagnosis [165].

#### 4.2.2. miRNA in HNC Metastasis

The main use of CTCs and ctDNA in the diagnosis and assessment of the metastatic potential of cancer, described earlier, can be supported and supplemented by the analysis of miRNA as a biomarker of these processes. A number of microRNA types have been associated with both an increase in the metastatic activity of the primary tumor, leading to the initiation of invasion of other tissues, as well as with the formation of a secondary tumor already in progress. The miR-21 mentioned in the previous section is associated not only with the diagnosis itself but also with the risk of metastases in laryngeal squamous cell carcinoma and oral squamous cell carcinoma [148,166]. Li et al. reported that increased miR-93 levels modulate the risk of lymph node involvement in HNSCC [167], and Fang et al. add that by promoting angiogenesis, miR-93 is responsible for tumor progression and increased invasive potential and consequently metastasis [168]. Both Sun et al. and Lu et al. indicate that reduced miR-9 levels in the plasma of patients with nasopharyngeal carcinoma are associated with a higher incidence of distant metastases [169,170], and these results were later confirmed for HNSCC by Hersi et al., who additionally suggested a mechanism of action and a CXCR4-specific inhibitor plerixafor as a potential therapeutic agent [171]. A tumor suppressor, miR-200a, has been extensively studied, and it has been revealed that its upregulation has been associated with a reduced risk of metastasis in OSCC, while downregulation leads to a significant increase in risk through targeting MYH10 to regulate the migration and invasion of nasopharyngeal carcinoma (NPC) cells [172,173]. Another potential explanation for the reduction in metastatic potential by miR-200a is the mechanism of making HNC cells more sensitive to ferroptosis, a form of cell death induced by iron-dependent accumulation of lethal lipid peroxidation [174]. Moreover, miR-200a takes a part in smoking-dependent carcinogenesis and metastasis resulting from DNA methylation that leads to progressive accumulation of nucleic acid damages in OSCC [175,176]. However, the reports are not in all cases equally consistent and harmonious. The latest reports on miR-134, which in recent years has been considered a factor increasing the risk of metastasis in HNC through targeting the WWOX [154] gene, are contradictory. Farag et al. proves that elevated levels of miR-134 increase the growth and migration of tumor cells by reducing the E-cadherin expression and, further, interacting with Programmed Cell Death 7 in OSCC [177], while Zhou et al. claims that miR-134 inhibits tumor stem cell migration and invasion in OSCC via downregulation of the PI3K-Akt signaling pathway by inhibiting LAMC2 expression [178]. This clearly indicates the need for further research in this area and a revision of existing knowledge in order to establish undeniable links between changes in the expression of individual miRNAs and the risk of metastasis.

#### 4.2.3. miRNA in Tumor Resistance to Radio and Chemotherapy HNC Treatment

Radiotherapy and chemotherapy are often the methods of choice for the treatment of HNC. Due to the great variety of cancer types falling into the head and neck group and the differences in location, the outcome of such treatment can be very difficult to predict [179,180]. In addition, even for the same cancer subtype, due to potential differences in the molecular profile, the effectiveness of treatment can vary drastically between patients [181]. Therefore, there is a need for patient profiling, which will allow not only the selection of an appropriate therapeutic method but also, and perhaps above all, patients to avoid ineffective treatment and the resulting very severe side effects. It is suspected that the mechanism behind miRNA modulation of resistance to chemo/radiotherapy is the effect on DNA repair systems. Since most of the cytostatic and cytotoxic drugs used in oncology are based on damage to the DNA of cancer cells, upregulation/downregulation of miRNAs affecting the effectiveness of these mechanisms will have an impact on the cancer’s ability to survive therapy [182]. For many years, research has focused on, firstly, biomarkers that will predict the outcome of a specific treatment and, therefore, select the best therapeutic path, and, secondly, biomarkers evaluated after treatment as an aid in assessing its effects. Both approaches have been successful, and a number of miRNAs have been used as biomarkers. Parwez et al. indicate that miR-15b-5p represents a potentially helpful biomarker for individualized treatment decisions concerning the management of HNSCC patients after examining relapse in HNC patients treated with intensity-modulated radiotherapy [183]. Zhao et al. suggest that miR-1278 inhibits the chemoresistance to cisplatin of nasopharyngeal carcinoma cells and might function as a novel therapeutic target in NPC treatment [184]. Nakashima et al. found that circulating miRNA-1290 is a potential biomarker for responding to chemoradiotherapy in patients with advanced OSCC [185]. Again, for OSCC, Shi et al. studied two miRNAs simultaneously and reported that miR-626 and miR-5100 are promising prognosis predictors for patients treated with adjuvant chemoradiotherapy [186]. This study is in line with the latest trend, which no longer uses single miRNAs but pairs or even entire sets. Such panels allow for a more comprehensive assessment of the patient’s condition and seem to be better predictors than single molecules. Hess et al. proposed the use of a five-miRNA signature composed of hsa-let-7g-3p, hsa-miR-6508-5p, hsa-miR-210-5p, hsa-miR-4306 and hsa-miR-7161-3p in the evaluation of HNSCC patient prognosis. The results indicate that this miRNA signature is a strong and independent prognostic factor for disease recurrence and survival of patients with HPV-negative HNSCC after adjuvant radiochemotherapy treatment [187]. Furthermore, for HNSCC, Chen et al. used a different 5-miRNA signature including miR-99a, miR-31, miR-410, miR-424 and miR-495 and found this set to be useful in predicting radiotherapy response [188]. Another interesting trend is an attempt to assess the effectiveness of radiotherapy after its application, as a support in evaluating the outcome of treatment. Here Pasi et al. indicate the post-radiation analysis of the miR-21, miR-191 and miR-421 set as an effective prognostic marker after radiotherapy [189].

#### 4.2.4. miRNA in General Prognosis of the HNC Outcome

The previously popular attempt to assess the general prognosis of HNC using miRNAs as biomarkers seems to have lost momentum in recent years. With more detailed tests and linking miRNA with the assessment of metastatic potential or the effectiveness of radiochemotherapy, the general assessment will probably not seem so attractive anymore. After all, it is affected by all the components mentioned, so by having the opportunity to evaluate individual elements and draw final conclusions on this basis, such a path seems to be more attractive. However research in this area is still being conducted, Panvongsa et al. indicates that plasma extracellular vesicle microRNA-491-5p may be a prognostic marker in the case of HNSCC [190], and Wu et al. identified seven miRNAs (hsa-miR-499a-5p, hsa-miR-99a-5p, hsa-miR-337-3p, hsa-miR-4746-5p, hsa-miR-432-5p, hsa-miR-142-3p, hsa-miR-137-3p) that are independent prognostic factors of HNSCC [191]. However, research in this area seems to specialize and, instead of general prognoses, focus on, for example, the risk of recurrence. Vo et al. indicates that miR-125a-5p is a marker of locoregional recurrence in HNSCC [192], while Rajan et al. go a step further and analyze the expression ratio of miR-196a to miR-204, which turns out to be a very strong predictor of disease recurrence in OSCC [193]. Ultimately, more and more advanced models are created that cover an ever-wider spectrum of elements for risk estimation and outcome prediction. Liu et al. proposed a 26-miRNA panel to assess the risk of developing OSCC in HPV-positive patients and explored the possibilities of using this panel to profile patients and determine personalized therapy [194].

### 4.3. lncRNA

Over 80% of human DNA is actively transcribed [195], and the RNA produced in this process serves both protein production and regulatory purposes. Only a small fraction of the RNA produced during transcription is then used for translation and protein production, while the vast majority (over 95%) is classified as non-coding RNA (ncRNA) [196]. Among this group, the majority are large particles (above 200 kDa). With the development of research and knowledge in this area, it turned out that the initial hypothesis that lncRNA is a remnant of the evolutionary process that does not perform any significant function is incorrect. Non-coding RNAs have diverse functions in the regulation of gene expression and cellular processes and regulate gene expression by acting as molecular scaffolds, competing with microRNAs for target sites or serving as decoys for transcriptional repressors. They can also interact with chromatin-modifying complexes, leading to changes in gene expression. Additionally, lncRNAs have been shown to have a direct role in cellular processes such as cell proliferation, apoptosis and differentiation [197,198,199,200]. With such a large impact on a wide spectrum of cellular processes, lncRNAs undoubtedly have an impact on the pathogenesis and development of several dysfunctions, and so far, have been implicated in a variety of diseases, including cancer [201], cardiovascular diseases [202,203] and neurodegenerative disorders [204,205]. In cancer, lncRNAs have been shown to play a role in tumor progression [206], invasion and metastasis [207] as well as in the regulation of drug resistance [208].

So far, lncRNA has received neither the attention nor the amount of research comparable to miRNA in head and neck cancers, and the results obtained are often incomplete and even ambiguous. The best-studied lncRNAs with oncogenic potential so far are lncRNA HOX transcript antisense RNA (HOTAIR), the elevated level of which was observed in OSCC [209] and is suspected of promoting metastasis in HNSCC [210], lncRNA metastasis–associated lung adenocarcinoma transcript 1 (MALAT1) with elevated levels in HNSCC [211], and lncRNA urothelial carcinoma antigen 1 (UCA1), which was elevated in tongue squamous cell carcinoma (TSCC) [211]. One of the suspected mechanisms of action is DNA methylation dysfunction (responsible for, inter alia, HOTAIR), and pattern analysis allows the use of lncRNA as a predictive indicator for HNSCC treatment outcome [212].

Finally, as a very interesting issue and an indicator of the complexity of lncRNA action, the potential miRNA/lncRNA interaction should be mentioned. lncRNA, which can be treated in this case as competitive endogenous RNA, could bind certain miRNAs and therefore isolate it from its target and in consequence promote HNSCC metastasis [213].

### 4.4. snRNA

Small nuclear RNAs are a class of short, non-coding RNA molecules that are involved in RNA metabolism in eukaryotic cells. They are medium-sized (100–200 nucleotides), stem-loop formations, stabilized by base-pairing interactions and have a highly conserved secondary structure [214]. snRNAs are usually associated with proteins to form small nuclear ribonucleoprotein particles (snRNPs). The main function of snRNAs is to participate in pre-mRNA splicing, where, along with other spliceosomal components, they recognize specific RNA sequences within pre-mRNA and catalyze the splicing reaction. In addition to their role in splicing, snRNAs have been implicated in other RNA processing and regulatory processes, including the 3’ end processing of histone pre-mRNAs, the processing of small nucleolar RNAs (snoRNAs) and the regulation of gene expression [215,216].

Dysregulation of snRNAs has been linked to various diseases, including cancer, where snRNA expression can be altered, leading to changes in pre-mRNA splicing patterns and the generation of aberrantly spliced isoforms that contribute to tumor development and progression. Additionally, snRNAs themselves can act as oncogenes or tumor suppressors. U1 snRNA, which is involved in the recognition of 5’ splice sites during pre-mRNA splicing, has been shown to regulate the expression of the oncogene MYC, and its upregulation is associated with poor prognosis [217]. U2 snRNA, involved in the recognition of the branch site during pre-mRNA splicing, when downregulated in lung cancer has been associated with poor prognosis [218].

Although there are no reports on the direct use of snRNA in HNC liquid biopsy, these molecules are used in the diagnosis of other types of cancer. RNU2-1f is an indicator in cholangiocarcinoma [219], SNORA25 in pancreatic cancer [220], RNU6 in breast cancer [221] and RNU2-1f in ovarian cancer [222]. Similar potential exists for HNSCC, where aberrant splicing events have been observed, leading to the generation of mRNA isoforms that may contribute to tumor progression [223]. In addition, the involvement of snRNA in the process of alternative transcription, the dysregulation of which may be a carcinogenic factor in HNSCC, indicates the potential diagnostic value of these molecules [224,225].

### 4.5. snoRNA

Small nucleolar RNAs are a class of small RNA molecules that primarily reside in the nucleolus and play essential roles in the chemical modification and processing of other RNA molecules, particularly ribosomal RNAs (rRNAs) and small nuclear RNAs. There are two main classes of snoRNAs: box C/D snoRNAs, which guide the site-specific 2’-O-methylation of rRNAs, and box H/ACA snoRNAs that are responsible for the pseudouridylation of rRNAs and snRNAs [226,227]. Both classes function by forming RNA-protein complexes known as snoRNPs (snoRNA-protein complexes), which consist of a snoRNA and associated proteins. The proteins within snoRNPs provide the catalytic activity required for the modification or processing of target RNAs, while the snoRNA provides the guide sequence that directs the modification to specific sites. Beyond their canonical role in ribosome biogenesis, snoRNAs have also been implicated in other cellular processes, including alternative splicing, mRNA stability and the regulation of gene expression [228,229].

Since snoRNAs are stable in body fluids and can be detected in circulating RNA fractions, they have gained attention as attractive candidates for liquid biopsy applications. Several studies have shown the potential of snoRNAs as diagnostic and prognostic biomarkers in colorectal cancer [230], non–small-cell lung cancer [231,232], ovarian cancer [233] and renal clear cell carcinoma [234]. In HNC, Xing et al. have shown that expression scoring of a small-nucleolar-RNA signature can serve as a prognostic predictor for HNSCC [235]. Studies showed that out of 113 survival-related snoRNAs, a five-snoRNA signature predicted prognosis with high sensitivity and specificity. Moreover, coexpression analysis revealed the five-snoRNA are involved in regulating malignant phenotype and DNA/RNA editing. Authors advocate that this five-snoRNA signature is not only a promising predictor of prognosis and survival but also a potential biomarker for patient stratification management. Similarly, Zou et al. reported 33 significantly dysregulated small nucleolar RNAs in HNSCC, of which down-regulated snoRNA expression levels correlated significantly with overall patient survival [236]. Finally, Petronacci et al. reported that in the case of OSCC they found 16 deregulated snoRNAs, of which one was over-expressed and 15 were under-expressed [237].

### 4.6. circRNA

Circular RNAs are a class of non-coding RNAs that form covalently closed loops by back-splicing, where a downstream splice donor site is joined with an upstream splice acceptor site. The lack of free ends makes circRNA resistant to exonucleases, making them highly stable and abundant in cells and in the bloodstream [238]. CircRNAs are generated from exons, introns or both and are usually tissue specific. They can act as microRNA sponges, interact with RNA-binding proteins and modulate transcription, and numerous reports indicated that they are capable of encoding functional peptides or proteins [239,240]. The ability to act as microRNA sponges or compete with endogenous RNAs (ceRNAs) allows circRNAs to prevent miRNA-mRNA interaction and thereby indirectly regulate gene expression [241,242]. Those features leading to the investigation of circRNA’s role in various biological processes include development, cell proliferation, differentiation and disease pathogenesis. Dysregulation of circRNAs has been observed in several diseases, including cardiovascular diseases, neurological disorders and nearly all types of cancers: gastric cancer [243], colorectal cancer [244], gastrointestinal cancer [245], breast cancer [246] and lung cancer [247]. Several distinct advantages of circRNA over linear RNAs as disease biomarkers seem to be the most important feature of circRNA. Firstly, circRNA is more stable than linear RNAs due to its closed-loop structure; secondly, circRNAs are abundantly expressed in almost all tissue samples; and, thirdly, circRNAs are expressed in a tissue-specific and development-stage-specific manner.

In the case of HNC, circRNA has been relatively extensively studied as a potential biomarker for liquid biopsy diagnostics. The wide availability and diversity of tissue-specific circRNAs (Lu et al. have isolated and identified 21444 distinct circRNA from laryngeal squamous cell carcinoma [248]) has led to great interest and intensification of research. Nath et al. reported nine different circRNAs dysregulated in HNSCC tumors that may serve as potential prognostic markers of HNC [249]. Similar findings have been reported for Laryngeal Squamous Cell Carcinoma [250,251], Tongue Cancer [252], Nasopharyngeal Carcinoma [253,254], Hypopharyngeal Squamous Cell Carcinoma [255] and Oral Squamous Cell Carcinoma [256,257]. Particular attention was paid to the influence of circRNAs on the proliferation process, where multiple circRNAs have been demonstrated to be involved in HNC development through the regulation of the proliferation process. CircRNAs are known to regulate proliferation by interfering with PI3K/Akt/mTOR, insulin-like growth factor, transforming growth factor (TGF) β, p53 signaling pathways and apoptosis [258,259,260,261,262]. Further studies also showed the usefulness of circRNA as a marker of invasion, metastasis and treatment prognosis, both in the case of chemotherapeutic and radiation therapy [263,264,265,266]. Finally, one of the most interesting mechanisms of circRNA pro-oncogenic activity is to increase the glucose uptake rate of tumor cells, thus providing fuel to meet their high metabolic demands through GLUT1 modification [267] and up-regulation of hexokinase 2 [268], both in OSCC.

All the aforementioned intensive research and the attention gained by circRNA in relation to its role in HNC make it undoubtedly the second most popular, after microRNA, RNA molecule considered as a biomarker of the pathogenesis of head and neck cancers. Although miRNA is in the field of liquid biopsy at a much more advanced stage of research, circRNA is also considered an excellent biomarker, and the panels created as proposals for faster diagnostics [249] give hope for the usefulness of these molecules.

### 4.7. piRNA

Piwi-interacting RNAs are a class of small non-coding RNA molecules, typically 24–32 nucleotides in length, that are primarily expressed in the germline cells of animals. They associate with Piwi proteins, a subfamily of Argonaute proteins, to form piRNA-induced silencing complexes (piRISCs) [269,270]. The main function of piRNAs is to protect the integrity of the germline genome by silencing DNA sequences capable of moving within the genome called transposable elements (TEs). Protection is achieved through a process known as piRNA-guided RNA cleavage, where piRNAs target and cleave complementary TE transcripts, thereby preventing their mobilization and potentially harmful effects on the genome [271]. In addition to TE silencing, piRNAs are also involved in gene regulation, epigenetic regulation and the maintenance of genome stability. Moreover, piRNAs are believed to play a role in chromatin remodeling, DNA methylation and the establishment of epigenetic marks in the germline [272]. While the majority of piRNAs are expressed in the germline, recent studies have suggested that piRNAs may also function in somatic cells and are associated with various diseases, including cancer. Mechanisms that are suspected as those by which piRNA may be involved in carcinogenesis include TE regulation, which are known to play a role in genomic instability and cancer development; epigenetic modifications, including DNA methylation and histone modifications; and gene expression modulation by targeting mRNA transcripts for degradation or translational repression [229]. Moreover, several studies have identified piRNAs that are specifically expressed in cancer cells or tumor tissues. These cancer-specific piRNAs may serve as diagnostic markers or therapeutic targets in gastrointestinal cancer [273], colorectal cancer [274,275], gastric cancer [276], breast cancer [277] or prostate cancer [278].

In the case of HNC, studies indicate the possibility of using piRNA as a biomarker in HNSCC, where the expression of specific piRNAs is deregulated and changes with both HPV status and type. A five-piRNA signature is able to delineate a subset of HPV-positive HNSCC patients with poor outcomes, highlighting the potential utility of piRNAs in patient management [279]. Zhou et al. reported a possible use of piRNA as a biomarker in OSCC where P-element Induced WImpy protein-like RNA-mediated gene silencing 2 (PIWIL2) regulates tumor cell progression, apoptosis and metastasis [280]. Li et al. found that in OSCC piwi-Interacting RNA1037 enhances chemoresistance and motility [281].

## 5. Summary and Future Perspectives

In a reality where, despite the allocation of huge financial resources for research and treatment, both the incidence of head and neck cancers and the mortality caused by them are increasing, there is an urgent need for innovative diagnostic methods. Since HNCs, like many cancers, are asymptomatic in their early stages, making them difficult to detect through conventional methods, the development of new diagnostic approaches that can identify cancer at its earliest and most treatable stages is crucial. Secondly, current diagnostic methods, such as tissue biopsies, can be invasive and uncomfortable and carry risks for patients. There is a need for non-invasive or minimally invasive alternatives that can reduce patient discomfort and enable repeated sampling for monitoring purposes and be accessible to a broader population. Moreover, HNC is a complex and heterogeneous disease, with significant variability in genetic and molecular characteristics among different individuals. Traditional diagnostic methods often provide limited information about heterogeneity, which can impact treatment decisions and patient outcomes. Therefore, there is a need for diagnostic methods that can capture the dynamic nature of cancer and provide comprehensive molecular profiling. In this context, liquid biopsy has emerged as a promising direction of research. As described in this article it offers a non-invasive approach to obtain tumor-derived biomarkers from readily accessible body fluids, such as blood. This eliminates the need for invasive procedures like surgical biopsies, making it safer, more convenient and potentially more widely applicable. Also, liquid biopsy allows for the analysis of various biomarkers, such as circulating tumor cells, circulating tumor DNA, tumor-derived extracellular vesicles and above all, as discussed in detail in this work, a whole variety of RNAs. These biomarkers carry genetic and molecular information derived directly from tumor cells, providing insights into tumor characteristics, genetic alterations and treatment response. By capturing the dynamic changes in these biomarkers over time, liquid biopsy can potentially monitor disease progression, detect minimal residual disease and identify acquired resistance mechanisms. Additionally, liquid biopsy has the potential to overcome tumor heterogeneity challenges by capturing genetic and molecular information from different regions of the tumor. This comprehensive molecular profiling can aid in personalized treatment decision-making and improve patient outcomes. However, despite the promising course and immeasurable benefits resulting from innovative medical methods, it should not be forgotten that the process of introducing new diagnostic or therapeutic approaches must be preceded by a long-term detailed research and implementation process that leaves no doubt as to the effectiveness and specificity of the new methods and, most importantly, the standardization of the procedures [282].

Have we already reached this point with liquid biopsy in HNC diagnostics? Definitely not; despite the high level of advancement of work, which is confirmed by the list of clinical trials for miRNAs (Table 2), it is still necessary to establish undeniable connections between the occurrence or changes in the level of individual molecules and a given clinical condition. Despite the plethora of studies and reports on the use of different types of RNA described in this article, in reality, all the assumed criteria for liquid biopsy are met by very few. Table 3 summarizes studies whose authors performed a study on patients with one of the histopathologically confirmed types of HNC, used only biological material collected in a non-invasive manner, qualifying such a procedure for liquid biopsy (saliva, blood, or blood derivatives), and then selected one or more biomarkers to statistically significantly differentiate the group with cancer from healthy controls.

The aforementioned lack of standardization of procedures can be observed, which seems to be the main problem which liquid biopsy is currently facing and the one that needs to be overcome if this procedure is to become a diagnostic standard.

Considering the history of the use of the particular types of RNA described in this manuscript, where the first “breakthrough” was mRNA, then the currently best-studied miRNA, and on the horizon, there are other promising targets in the form of snRNA, snoRNA or piRNA, but it is currently difficult to say which of the discussed molecules will become the standard in the future. Perhaps it will not even be one of them alone, but an intermolecular panel, given the increasing reports of interactions and joint influence of different types of RNA. One thing is certain, however—the potential benefits fully justify the pursuit of finding this RNA, which will allow for the widespread introduction of liquid biopsy into HNC diagnostics.

## Figures and Tables

**Figure 1 cells-12-01916-f001:**
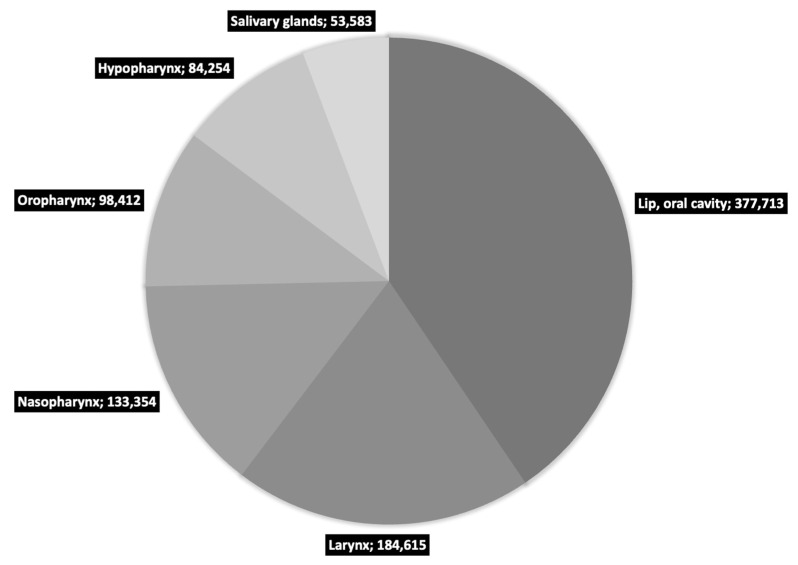
Number of new HNC cases in 2020 worldwide as reported by GLOBOCAN [1].

**Figure 2 cells-12-01916-f002:**
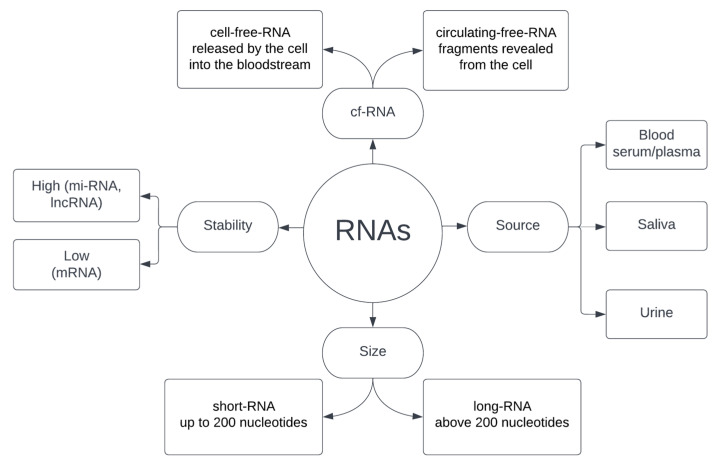
Characteristics of RNA used in liquid biopsy.

**Table 1 cells-12-01916-t001:** RNA subtypes used in liquid biopsy [98,99,100,101,102,103,104,105,106].

Subtype	Code	Characteristics
messenger RNA	mRNA	fragmented into pieces of different lengths; unstable; low occurrence
microRNA	miRNA	high occurrence; high specificity; length of about 22 nucleotides; high stability
long non-coding RNA	lncRNA	more than 200 nucleotides; high stability resulting from the presence of extensive secondary structures; high specificity expression pattern
small nuclear RNA	snRNA	about 150 nucleotides; associated with a set of specific proteins
small nucleolar RNA	snoRNA	up to 300 nucleotides; stable
circular RNA	circRNA	covalently closed continuous loop; abundant; average length 550 nucleotides
piwi-interacting RNA	piRNA	largest class of small non-coding RNA; 26–31 nucleotides

**Table 2 cells-12-01916-t002:** Clinical trials using miRNA in HNC reported by clinicaltrials.gov (April 2023).

Study Title	Conditions	Ref.
MicroRNA Markers in Head and Neck Cancers	Squamous Cell Carcinoma of Head and Neck	[144]
Tertiary Prevention of Head and Neck Cancer With a Dietary Intervention	Cancer of Head and Neck	[145]
Hemopurifier Plus Pembrolizumab in Head and Neck Cancer	Squamous Cell Carcinoma of the Head and Neck	[146]
Neoadjuvant Nivolumab for Oral Cancer Combined With FDG and Anti-PD-L1 PET/CT Imaging for Response Prediction	Oral Cavity Squamous Cell Carcinoma	[147]

**Table 3 cells-12-01916-t003:** Summary of the different types of RNA that can act as a biomarker in liquid biopsy in HNC.

RNA Type	Subjects	Cancer	Sample	Detection Method	Findings	Ref.
mRNA	78	OSCC	Serum and saliva	Western Blotting/RealTime PCR	Transgelin expression higher in patients’ saliva (*p* < 0.01) but not serum (*p* < 0.05)	[129]
mRNA	33	OSCC	Saliva	RealTime PCR	MAOB–NAB2 is predictive of OSCC (AUC, 0.91; sensitivity, 0.92; and specificity, 0.86)	[131]
mRNA	32	OSCC	Saliva	RealTime PCR	IL8, IL1B, DUSP1, HA3, OAZ1, S100P, and SAT exhibited at least a 3,5-fold elevation in OSCC saliva (*p* < 0.01)	[132,133]
miRNA	96	HNSCC	Blood	Western Blotting/RealTime PCR/Immunohistochemistry	miR-134 expression value had a predictive power of 0.73 for distinguishing malignant from nonmalignant states	[154]
miRNA	106	HNSCC	Plasma	RealTime PCR	In NPC serum samples, miR-762 was significantly upregulated (*p* < 0.001)	[158]
miRNA	216	HNC	Blood	RealTime PCR	let-7 microRNA binding site variant in KRAS is associated with lymph node metastasis [OR (%95 CI) = 2370 (1.03–5.41), *p* = 0.03, χ2 = 4.38]	[160]
miRNA	103	HNSCC	Blood	RealTime PCR	miR-93 overexpression was associated with tumor progression, metastasis and poor prognosis (*p* < 0.05)	[167,168]
miRNA	294	NPC	Blood	RealTime PCR	Low level of plasma miR-9 was correlated with lymphatic invasion and advanced TNM stage (*p* < 0.05)	[169]
miRNA	37	OSCC	Saliva	RealTime PCR	miRNA-200a and miRNA-134 were upregulated in cancer patients (*p* < 0.00001)	[177]
miRNA	10	OSCC	Plasma	RealTime PCR	Expression level of miR-1290 was significantly lower in the plasma of patients (*p* < 0.01)	[185]
miRNA	73	HNSCC	Plasma	RealTime PCR	miR-491-5p is prognostic indicator for overall survival ([HR] 5.66, 95% confidence interval, 1.77–18.01; *p* = 0.003)	[190]
lncRNA	28	OSCC	Plasma	RNA sequencing	lncRNA HOX transcript antisense RNA (HOTAIR) levels were elevated in OSCC (*p* < 0.05)	[209]
snoRNA	8	OSCC	Plasma/Tissue	Affymetrix miRNA 4.1 Array Plate microarray platform	16 deregulated snoRNAs (1 overexpressed and 15 underexpressed, fold change above 2.01, *p* < 0.05)	[237]
circRNA	100	NPC	Blood	RealTime PCR	Expression of hsa_circRNA_001387 was significantly elevated in patients with lymph node metastasis and distant metastases (*p* < 0.01)	[263]

## Data Availability

No new data were created, all data are available on request from the authors of the cited articles.

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
