# Peer review of "RNA-Based Liquid Biopsy in Head and Neck Cancer"

_cells, 2023, doi:10.3390/cells12141916_

Round 1

Reviewer 1 Report

There is a lot of speculation expressed in this article, could be stronger in terms of pragmatic interpretation of published data.  The article brings together a great deal of published information which is useful in providing a perspective.  There is no mention of the potential of metabolomics research.

The main question being addressed is utility of RNA as "liquid biopsy" in head and neck cancer. This is a good review of the potential use of blood-based markers including circulating tumour cells, DNA and RNA, based on the literature. This is not original research but a review of published material. It is useful as a compilation of work done by various investigators, well referenced.

In terms of improvement:

1) the review mentions the issue of various types of head and neck cancer - salivary gland, squamous cancers of various types, p16-positivity (HPV-related) and nasopharyngeal cancer; however, the review doesn't address the differences in use of RNA-based biopsy between these sub-types.

2) the review speaks to results of "liquid biopsy" in other types of cancer outside of the head and neck (which is the topic of this manuscript). This information isn't really relevant to this question.

3) There is no mention of metabolomic markers, which is relevant to this review. There has been significant work in this area and I think there should be a section added relating to metabolomics for completeness.

In terms of methodology and controls, these questions are not relevant to this type of a review article. The conclusions appear appropriate and relevant. The tables and figures appear O.K.

This is a compilation of information gleaned from the literature.  It is useful mainly as an overview in providing a perspective on this complex and expanding field.  Unfortunately there is very little new information presented in this publication.  There are some minor editorial corrections needed such as in line 233 where the word "secret" should be "secretion", line 249 should be "under-60 years of age group".  Line 268 - 270 relating to miRNA doesn't seem to make sense, line 330 "overregulation" should be "upregulation".  In line 422 I would use "lncRNA" rather than "ncRNA" to avoid confusion.  A section on metabolomics should be added for completeness.

Author Response

Dear reviewer,

we sincerely thank you for your valuable comments on our manuscript, we have read them with care and believe that their implementation will make our article better. Regarding the details of the corrections which, according to your suggestions, have been made, we inform:

- regarding the first point, a similar objection was raised by the second reviewer and as suggested by him we have added a table (Table 3) which summarizes the use of RNA for liquid biopsy in different types of HNC

- regarding point two, we have truncated the information on cancer types other than HNC, however, we firmly believe that it should not be removed altogether as it puts HNC in the context of research in other cancer types - overly often what has worked well in one type of cancer is tested in other types - this may be a clue to readers in which direction to potentially expect further HNC research to go

- regarding point three, we completely agree that the chapter on metabolic biomarkers is relevant to this work, therefore we have added the subsection "2.4. Metabolomic markers"

Regarding the minor editorial corrections that were mentioned - all have been corrected

We firmly believe that you will now find the article much better.

Thank you,

The authors.

Reviewer 2 Report

The authors describe the knowledge gained on RNA detection in liquid biopsy for cancer detection and monitoring, particularly in head and neck cancer.  

Comments

1 - The authors describe the experimental results related to the determination of RNAs in liquid samples of cancer patients. For head and neck cancer studies only and for each RNA class, the sample type, statistically significant results, series consistency, detection method, and experimental design should be summarized in a table, to give readers a concise representation of the current state of research.

2- The text offers an overview of the opportunities that the different classes of RNAs offer for the detection and monitoring of cancer. The authors should discuss the convergences and divergences that emerge from the comparison of the results reported in the new Table.

Minor

- the acronyms definition is repeated

- line 133, correct Individualised.

- lines 220-224, the sentence is repeated.

- line 307, correct Al.

The quality of the English language is adequate. 

Author Response

Dear reviewer,

we sincerely thank you for your valuable comments on our manuscript, we have read them with care and believe that their implementation will make our article better. Regarding the details of the corrections which, according to your suggestions, have been made, we inform:

- regarding comments 1 and 2, we created a table (Table 3) as suggested, included the mentioned information in it, and then discussed the data contained therein

Regarding the minor editorial corrections that were mentioned - all have been corrected

We firmly believe that you will now find the article much better.

Thank you,

The authors.

Round 2

Reviewer 1 Report

I believe all the issues identified in the original review have been very satisfactorily addressed.  Thank you, well done.

Reviewer 2 Report

The synthesis of the results obtained in sixteen studies on the detection of RNAs in liquid samples of HNC patients clearly highlights the lack of reproducibility. The authors suggest the standardization flaw as the main limitation. The review leads us to the conclusion that standardization of procedures as well as reliable and translational biomarkers can only emerge in (necessarily collaborative) studies of thousands of samples from well categorised HNC patients detecting all different types of RNA.

lines 324-328, the two sentences are repeated differently in some words.

Adequate.